# 'This isn't what mine looked like': a qualitative study of symptom appraisal and help seeking in people recently diagnosed with melanoma

Fiona M Walter,[1,2] Linda Birt,[1] Debbie Cavers,[3] Suzanne Scott,[4] Jon Emery,[1,2] Nigel Burrows,[5] Gina Cavanagh,[6] Rona MacKie,[7] David Weller,[3] Christine Campbell[3]

For numbered affiliations see end of article.

**Correspondence to**
Dr Fiona M Walter;
fmw22@medschl.cam.ac.uk

## ABSTRACT

**Objective:** To explore symptom appraisal and help-seeking decisions among patients recently diagnosed with melanomas, and to compare experiences of people with 'thinner' (<1 mm) and 'thicker' (>2 mm) melanomas, as thickness at diagnosis is an important prognostic feature.

**Methods:** In-depth interviews with patients within 10 weeks of melanoma diagnosis explored the factors impacting on their pathways to diagnosis. Framework analysis, underpinned by the Model of Pathways to Treatment, was used to explore the data with particular focus on patients' beliefs and experiences, disease factors, and healthcare professional (HCP) influences.

**Results:** 63 patients were interviewed (29–93 years, 31 women, 30 thicker melanomas). All described their skin changes using rich lay vocabulary. Many included unassuming features such as 'just a little spot' as well as common features of changes in size, colour and shape. There appeared to be subtly different patterns of symptoms: descriptions of vertical growth, bleeding, oozing and itch were features of thicker melanomas irrespective of pathological type. Appraisal was influenced by explanations such as normal life changes, prior beliefs and whether skin changes matched known melanoma descriptions. Most decisions to seek help were triggered by common factors such as advice from family and friends. 11 patients reported previous reassurance about their skin changes by a HCP, with little guidance on monitoring change or when it would be appropriate to re-consult.

**Conclusions:** Patients diagnosed with both thinner and thicker melanomas often did not initially recognise or interpret their skin changes as warning signs or prompts to seek timely medical attention. The findings provide guidance for melanoma awareness campaigns on more appropriate images, helpful descriptive language and the need to stress the often apparently innocuous nature of potentially serious skin changes. The importance of appropriate advice, monitoring and safety-netting procedures by HCPs for people presenting with skin changes is also highlighted.

## Strengths and limitations of this study

- This study is the first exploration of symptom appraisal and help seeking among people diagnosed with 'thinner' melanomas (T1, very good prognosis, 5 year disease-free prospects 95%), compared with those with 'thicker' melanomas (T3 and T4, less good prognosis, 5 year disease-free prospects <55%).
- The study did not identify clear discriminating features in the diagnostic pathway, or features of thinner versus thicker melanomas.
- The findings highlight a mismatch between the information people need when assessing their skin changes and the information and images currently available, thus providing opportunities to incorporate more appropriate descriptive language, images and information into targeted community awareness campaigns as well as by the National Health Service (NHS) and charities via their websites and promotional materials.
- A small but important minority of participants did not have their developing melanomas recognised during their first primary care consultation, and were not provided with enough information about ongoing assessment of further skin changes or when to return to their clinician. These 'safety-netting' opportunities could be improved by more systematic approaches by healthcare professionals.
- Using semistructured interviews close to diagnosis allowed in-depth exploration of the participants' experiences and views, but the accounts are necessarily retrospective and subject to recall and framing bias.

## INTRODUCTION

Diagnosing melanoma earlier is high on the UK health policy agenda; it is estimated that around 190 deaths from melanoma could be avoided each year if survival rates in England matched the best in Europe.[1] Worldwide

melanoma incidence rates are increasing faster than any other solid tumour. In the UK the incidence has quadrupled since the 1970s[2]; similar incidence rises have been reported across Europe,[3] [4] the USA[5] and Australia.[6] In the UK there were more than 2209 deaths and 12 800 new cases diagnosed in 2011, with a disproportionately high rate among people aged less than 50 years.[2] The most important prognostic factor is the tumour thickness at diagnosis according to the Breslow scale (T classification).[7] Patients with a primary melanoma ≤1 mm at diagnosis (T1) currently have 5-year disease-free prospects of over 95%, while for tumours ≥2 mm at diagnosis this is lower, falling to <55% with lymph node involvement but no metastatic spread.[8] Tumour thickness is also associated with rapid growth which occurs more frequently in elderly men.[9]

Timely diagnosis can be influenced by the diagnostic skills of general practitioners (GPs). A recent analysis of the Cancer Patient Experiences Survey 2009 and the 2010 RCGP cancer audit data reported that more than 90% of people diagnosed with melanoma were seen by their GPs less than three times before diagnosis, compared with 60–80% for the majority of cancer types.[10] This suggests that most melanomas are recognised by GPs and appropriately referred to specialist care in England.

Timely diagnosis can also be influenced by people's symptom appraisal and help-seeking behaviour. Compared with other cancers, people with melanoma have among the longest time between first noticing a symptom and presenting to their GP,[11] [12] suggesting that the major opportunity to diagnose melanoma earlier is prompting earlier presentation to healthcare through signs and symptom awareness campaigns.[13] This requires an understanding of how people interpret changes in their moles or new lesions. We present findings from an in-depth interview study with UK patients recently diagnosed with 'thinner' (T1) compared with 'thicker' primary melanomas (T3 and T4), which aimed to explore the processes and experiences of symptom detection and help-seeking decisions leading to melanoma diagnosis.

## METHODS
### Design and ethics
Semistructured face-to-face in-depth interviews were conducted with adults diagnosed with invasive cutaneous melanoma within the previous 10 weeks.

### Setting and recruitment
Potential participants were identified and recruited by the melanoma/skin cancer nurse specialists via the weekly multidisciplinary team meetings of dermatologists, plastic surgeons and oncologists at two regional hospitals: Cambridge University Hospitals NHS Foundation Trust in the East of England, and the Edinburgh Royal Infirmary, NHS Lothian, Scotland.

These hospitals together serve a population of approximately 1.4 million, and the MDT meetings review more than 400 new cases of invasive cutaneous melanoma each year.

All adults aged 18 and above newly diagnosed with a primary invasive cutaneous melanoma (staged as ≤1 mm (T1, 'thinner') or ≥2 mm (T3 and T4, 'thicker') at the two participating hospitals were eligible for inclusion unless the melanoma/skin cancer nurse specialists felt that they were not suitable on clinical grounds (other severe physical or mental health conditions). Patients were mailed an invitation letter with a patient information sheet. As T3 and T4 melanomas are diagnosed at about 25% of the rate of T1 melanomas, we recruited all those with thicker melanomas who agreed to take part. At the same time we purposively sampled people with T1 melanomas by age, gender, location and season to ensure that we had a broad range of views and experiences, and we continued until saturation of data. Reasons for not selecting patients for interview included: sampling decisions (n=34), lost to follow-up (n=6) and ill health (n=1).

### Data collection
Interviews were undertaken between January 2012 and January 2013. In each area an experienced researcher used a semistructured approach with an interview schedule informed from the literature,[14] [15] our collective expertise from interviewing patients recently diagnosed with other cancers,[16] and a pilot study (n=17, conducted during the early stages of the study, and including patients interviewed >10 weeks post-diagnosis (n=12), or with melanoma histology which did not fit the inclusion criteria (n=5, Breslow thickness 1–2 mm or indeterminate)). The theoretical approach of the Model of Pathways to Treatment[17] [18] (figure 1) was used to underpin the interview schedule, exploring the processes that occurred within each time interval and focusing on: how initial symptoms were noticed; personal risk perceptions; the language used to describe symptoms and changes over time; the participant's decision-making and triggers to help seeking; and the experience of the diagnostic process of primary and secondary care from the patient perspective. A calendar-landmarking technique[19] was used as an adjunct to the interviews, to establish the timing and details of events which led to the melanoma diagnosis, together with diaries and letters that participants referred to during this process. Participants were also invited to make a pencil drawing/s of their skin cancer as it developed; ongoing analyses are examining perceptions of lesions over time and comparing the drawings with clinical images. At the end of each interview, participants completed a short questionnaire to provide demographic data and information about their skin and hair colour and their skin's response to ultraviolet (UV) light using the widely validated Fitzpatrick Scale.[20]

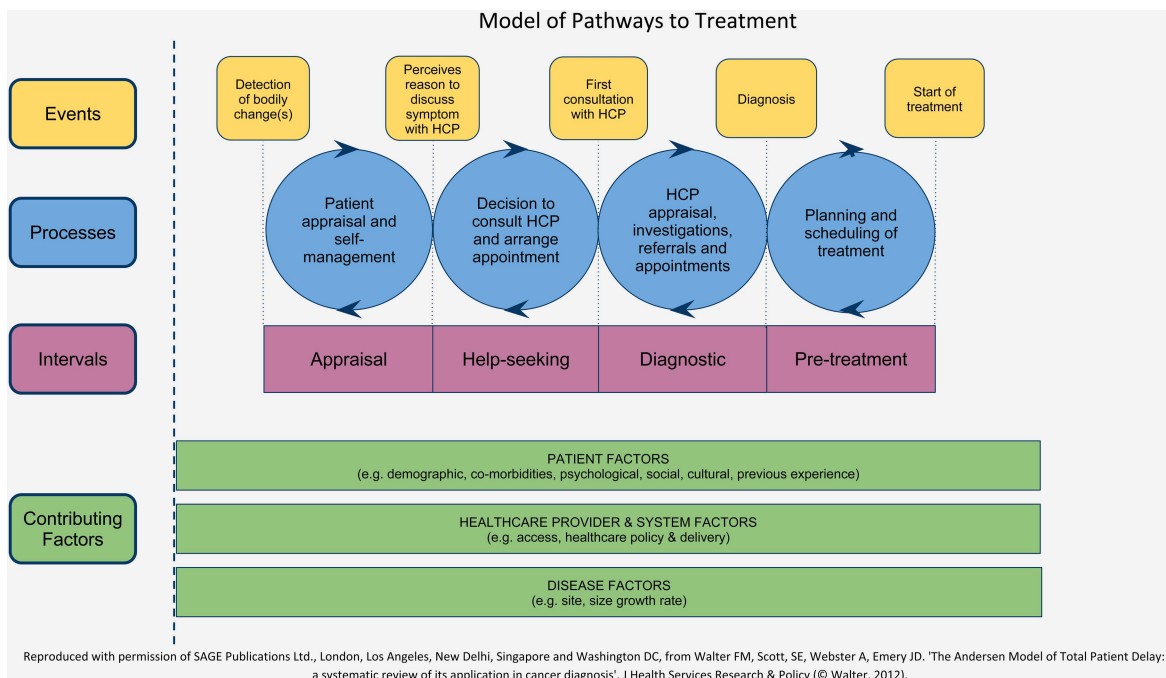

Reproduced with permission of SAGE Publications Ltd., London, Los Angeles, New Delhi, Singapore and Washington DC, from Walter FM, Scott, SE, Webster A, Emery JD. 'The Andersen Model of Total Patient Delay: a systematic review of its application in cancer diagnosis'. J Health Services Research & Policy (© Walter, 2012).

**Figure 1** Model of Pathways to Treatment (HCP, healthcare professional).

Interviews were undertaken as soon as possible after diagnosis, with all interviews completed within 10 weeks of diagnosis and the majority within 6 weeks. Interviews lasted between 40 and 65 min and were conducted primarily in the participant's home, although two people chose to be interviewed in university offices. Patients were sometimes accompanied by a family member, usually spouse or daughter. Audio-recordings of interviews were professionally transcribed verbatim and anonymised.

## Analysis

All interview transcripts were repeatedly read and re-read by the two researchers LB (nursing background) and DC (health services researcher), and the members of the 'core' analysis team also read the majority of the transcripts (FMW, academic GP; SS, health psychologist; CC, primary care researcher). Analysis was an iterative process starting near the beginning of data collection and using the 17 pilot interviews to develop our analytic strategy. We used the approach of Framework analysis to create and establish meaningful patterns in five phases, namely: familiarisation with the data, generating initial codes, inductively searching for themes among codes, index charting and mapping of data, before finally defining and naming themes.[21] The coding and data management were supported by NVivo software (QSR International, V.9). The Model of Pathways to Treatment (figure 1) was also used to underpin the analysis with a theoretic model for the different intervals and processes that occur along the pathway to diagnosis and treatment, in order to accurately assess the time intervals, their content and context. The final themes were agreed through a series of meetings involving all five 'core'

researchers, and a consensus meeting with the wider study team.

The analysis focused on the main themes within the time to presentation (TTP), defined as from the first detection of skin change to the first consultation with an HCP.[22 23] This interval comprises the appraisal and help-seeking intervals,[17 18] and the analysis examined patient and healthcare factors as well as 'disease' factors, relating to the developing melanoma. When the first consultation did not result in a referral, we also included further iterative processes until the next consultation in the analyses. Participants with shorter intervals tended to use diaries and have good recall of the relevant dates. People with longer intervals tended to have vaguer recollections, particularly around the time they had first detected any skin change. While participants were often able to discuss triggers to help seeking, they were less able to recall the precise dates of these triggers, and we therefore do not present the separate durations of the appraisal and help-seeking intervals.

We went on to examine our themes by comparing narratives from participants diagnosed with thinner and thicker melanomas and by the melanoma types within these groups. We further validated our themes by examining the whole dataset stratified by gender, age (less than 60 vs 60 and over, and 80 and over), educational level (no further education vs further education) and geographical location (Cambridge vs Edinburgh). Credibility was increased by the two researchers together undertaking coding and producing code tables throughout the analytic process, and reaching consensus from the potentially wide range of interpretations across the 'core' analysis team.

## RESULTS

A total of 241 adult patients were approached to take part in this study (Cambridge 114, Edinburgh 127), 121 were willing to participate (50%: Cambridge 53%, Edinburgh 47%), and 63 were interviewed.

### Patient characteristics

Table 1 shows the demographic and self-reported skin characteristics of the 63 study participants and the clinical characteristics of their melanomas, comparing participants with thinner (n=33, median Breslow thickness 0.5 mm, range 0.1–0.9 mm) and thicker (n=30, median

Breslow thickness 3.5 mm, range 2.1–12.0 mm) melanomas. While people with thinner melanomas were younger (60.5 vs 66.1 years), the groups were otherwise similar for sociodemographic factors. One quarter of the group reported a family history of melanoma, while eight participants reported previous skin cancer (melanoma 2, basal cell carcinoma 6); we were only able to verify the two melanomas with histology reports. The thinner melanomas were all histologically reported as superficial spreading melanomas (SSM) and lentigo maligna melanomas (LMM) apart from one diagnosed as part SSM and part NM (nodular melanomas; 'other').

**Table 1** Characteristics of study participants (n=63), and clinical characteristics of their melanomas, comparing thinner (n=33) and thicker (n=30) melanomas

| | Thinner melanomas (<1 mm Breslow thickness) | Thicker melanomas (>2 mm Breslow thickness) |
|---|---|---|
| Age at interview | | |
| Mean age±SD (range) | 60.5±14.6 (29–85) | 66.1±15.5 (36–93) |
| Less than 60 years (n=23) | 14 (61%) | 9 (39%) |
| 60 years and over (n=40) | 19 (47%) | 21 (53%) |
| Gender | | |
| Male | 14 (42%) | 17 (58%) |
| Female | 19 (58%) | 13 (42%) |
| Education | | |
| No further education | 21 (64%) | 21 (70%) |
| Further education | 12 (36%) | 9 (30%) |
| Fitzpatrick scale: skin colour* | | |
| Type I (white skin, very fair) | 7 (21%) | 2 (6%) |
| Type II (white skin, fair) | 6 (18%) | 11 (37%) |
| Type III (creamy white, any hair) | 17 (52%) | 16 (53%) |
| Type IV (brown, Mediterranean) | 3 (9%) | 1 (3%) |
| Fitzpatrick scale: skin reaction to sun* | | |
| Type I (always burns, never tans) | 4 (12%) | 3 (10%) |
| Type II (usually burns, tans with difficulty) | 11 (33%) | 10 (33%) |
| Type III (sometimes mild burn, gradually tans) | 10 (30%) | 11 (37%) |
| Type IV (rarely burns, tans easily) | 6 (18%) | 6 (20%) |
| Type V (very rarely burns, tans very easily) | 2 (7%) | 0 |
| Melanoma location | | |
| Head and neck | 6 (18%) | 9 (30%) |
| Trunk† | 9 (27%) | 4 (14%) |
| Upper limb | 10 (30%) | 7 (23%) |
| Lower limb | 8 (24%) | 10 (33%) |
| Melanoma type | | |
| Superficial spreading melanoma (SSM) | 25 (76%) | 10 (33%) |
| Nodular melanoma (NM) | 0 | 10 (33%) |
| Lentigo maligna melanoma (LMM) | 7 (21%) | 2 (7%) |
| Others‡ | 1 (3%) | 8 (27%) |
| Melanoma TNM stage | | |
| I§ | 33 (100%) | 0 |
| II¶ | 0 | 23 (77%) |
| III** | 0 | 7 (23%) |
| IV | 0 | 0 |

*Self-reported, not verified in medical records.
†Includes melanomas on the back (thinner 6, 18%; thicker 3, 10%).
‡Thinner: mixed type (SSM and NM)×1; thicker: LMM 2, acral 1, malignant blue naevus 1, unclassified 6.
§Stage IA=27, stage IB=6 (T1–2a, N0, M0).
¶Stage IIA=11, stage IIB=5, stage IIC=7 (T2b–4b, N0, M0).
**Stage IIIA=6, stage IIIB=1 (T1a–4a, N1a–2c, M0).
LMM, lentigo maligna melanomas; NM, nodular melanomas; SSM, superficial spreading melanomas; TNM, TNM Classification of Malignant Tumours (TNM system).

Owing to our sampling strategy there was a higher prevalence of nodular melanomas than in reported local figures. However, only a third of the thicker melanomas were NM (n=10), while a third were SSM (n=10), and the remaining third had 'other' diagnoses (LMM 2, acral 1, malignant blue naevus 1, unclassified 6). Of the nine participants diagnosed with melanoma on their back, seven were male, and three had thicker melanomas (NM 2, SSM 1).

## Duration of skin changes

Four participants (male 3, female 1) had their melanomas diagnosed opportunistically by an HCP (3 GPs, 1 oncologist); all these were thinner melanomas. The TTP was between 1 and 303 weeks (thinner: median TTP 21 weeks, range 1–303 weeks, 5 longer than 52 weeks; thicker: median TTP 19 weeks, range 1–156 weeks, 7 longer than 52 weeks). Most participants who presented with skin changes were referred after their first primary care consultation. The remainder were referred after their second consultation (n=11); none reported more than two consultations prior to referral. Comparisons between those with thinner (n=4) and thicker (n=7) melanomas who were referred after a second primary care consultation are presented in table 3 and discussed later (see section on Healthcare providers and system factors).

The main emerging themes within the appraisal and help-seeking intervals are discussed below. Throughout this section quotations are accompanied by information about gender (male (M), female (F)), age, melanoma group (thinner or thicker), type of melanoma (SSM, NM, LMM, other) and symptom duration as TTP in weeks (including first and second presentations).

## The appraisal interval

The 59 participants who detected their melanoma themselves described a variable and complex process of appraisal and reappraisal of their skin, against their background knowledge of 'normal skin changes' and potential risk factors. We found no evidence of differences between people with thinner and thicker melanomas across any of these themes.

### Patient factors
#### Explanations for skin changes

Awareness of a skin change, either a new lesion or a change in an existing lesion, did not usually cause any initial concern as it seemed so innocuous, and was often attributed to normal life changes such as pregnancy or aging.

> I didnt recognise it as something that was different, because Ive got quite a few moles on my skin so therefore I thought, Has this been here before, or am I just imagining that I havent seen it before? [F, 68, thinner, SSM, 52w]

> Perhaps because Id been pregnant and everything was darker anyway or you know, I didnt take any notice. [F, 36, thicker, NM, 17w]

Other explanations were also often made, such as an insect bite or injury when the participant had been outside or in the garden.

> Since Id been outside to a barbeque and I thought, oh well Ive been bitten, its just bitten there on the mole. [F, 54, thicker, SSM, 1w]

Skin changes were sometimes attributed to another skin condition (such as psoriasis) if it presented in a similar way; in these cases participants' previous experiences of a benign condition could influence their perception of the potential seriousness of the new skin changes.

### Prior beliefs about melanoma and its risk factors

Skin changes were appraised within the context of peoples' prior beliefs about melanoma and its risk factors, and their life experiences. Participants often used the terms skin cancer and melanoma interchangeably, and their prior awareness of melanoma varied widely. Whereas some participants noted that they had no awareness at all, others described gaining some knowledge about melanoma via TV programmes, magazines, the internet, and occasionally, health promotion material. A minority had heightened awareness through the melanoma experience of a family member or friend, or even a celebrity. A family history of melanoma or a personal previous melanoma led several people to have heightened risk perception and awareness and to quickly identify skin changes as a potential melanoma; all these people sought help rapidly and presented with thinner melanomas.

> I wouldnt have known what they were talking about. [M, 62, thicker, SSM, 52w]

> Because my mum has had a melanoma ten years ago so Ive always been aware to keep a check on my moles. [F, 29, thinner, SSM, 3w]

Many participants showed some understanding of the risk factors associated with melanoma and/or skin cancer when they discussed having lived in hot climates, or having suffered from sunburn, especially as a child. However, some were quite certain that they had never exposed themselves to the risk of UV damage.

> I thought I had been careful about sitting out in the sun. [F, 57, thicker, NM, 36w]

Prior knowledge or experience of melanoma and its risk factors did not appear to be related to educational levels, nor to melanoma thickness at diagnosis.

## Do skin changes 'match' a melanoma?

While some participants admitted to prior knowledge of the symptoms and signs of a melanoma, such as 'jagged edges' or change in colour, only a few people had known that an itchy or bleeding mole was a 'bad' sign. Only two people noticed a match between their observed skin changes and their mental image of a melanoma, and this match appeared to prompt appropriate help seeking, leading to shorter TTP.

> I dont know when I learnt it, but it was just in my subconscious that ooh I need to go and, its an itchy mole, thats not good. [M, 45, thicker, SSM, 4w/52w]

Strikingly, the majority of participants reported that their observed skin changes did not match their mental image (which had arisen from the melanoma experience of a family member or friend, from written and visual images, or from their knowledge of other cancers, see box 1). When the changes did not match their mental images, people appeared more likely to 'normalise' their skin changes, or adopt other explanations, thus delaying help seeking and diagnosis. Thus, the appraisal interval was often prolonged when there was a 'mismatch' between the mental image people had of melanoma and the way in which their own skin changes developed.

## Disease factors: skin changes

Most participants used rich and vivid lay vocabularies to describe their skin changes, for example, "like a black fly squashed on a mirror" (M, 48, thinner, SSM, 2w). Table 2 shows descriptions of skin changes noticed by participants, displayed according to the items of the Glasgow seven-point checklist (7PCL).[24] It also gives descriptions not commonly found on checklists. For instance, many people reported surprise at the small size of their melanoma, describing it as 'just a little spot'. Some also reported a 'spot on a mole', or that their skin change had been 'always there' or a 'new lesion'; a few reported their lesion as 'different to the others' (resonating with the Ugly Duckling sign[25]).

Overall, table 2 shows that thinner and thicker melanomas can show any of the changes described in the 7PCL. However, there is a suggestion of slightly different patterns. In particular, patients with thicker melanomas, both NMs and SSMs, described the so-called 'minor features' of bleeding, oozing and itch more often. They also described both horizontal and vertical growth, again, irrespective of pathological type. Patients with thinner lesions discussed changes in shape more often. We were not able to find any differences in descriptions of skin changes between gender, age, educational level or geographical region.

## The help-seeking interval

Reasons for waiting before seeking help included weighing up the priority of their skin change against other commitments. Many participants had been encouraged by other people to seek the advice of an HCP for their skin change. Emotions such as fear of a serious condition, cancer or treatment were seldom mentioned and seemed to play little part in most people's decision-making, either to promote or delay help seeking. More were concerned about going to see their GP with only minor symptoms, and wasting the GP's time.

---

**Box 1** Illustrative quotations of a 'mismatch' between observed skin changes and 'mental images' of a melanoma

**Comparison with experience of a family member's melanoma**
'My mother in law had skin cancer on her back, so I expected melanoma to be much bigger: my little mole was nothing.' (F, 53, thinner, SSM, 14w/52w)

**Comparison with information**
'I suppose from the descriptions that I've read about melanomas... it didn't ring any alarm bells... Okay, it has to start somewhere but, as it developed, (I expected) it would become more raised, it would be scaly and rough, it would be more inflamed-looking. But because this just remained completely flat on the skin...it didn't meet any profile that I was expecting'. (F, 63, thinner, LMM, 104w)

'It's not like a mole, you can see a mole changing colour or shape or texture, you can read all about that, but as my wife says, nothing in the leaflets says anything about under the nail'. (M, 60, thicker, other, 1w)

'Because melanomas, he says, are black. Now, this growth on my knee, it was just like a warty growth, with a scarlet top on it, and... there was no discolouration in it at all...I think that's maybe how a lot of folk'll not think of these things, because it doesn't look like what you think it's supposed to be, if you ken what I mean. It's just like a bit of skin rising up'. (M, 52, thicker, SSM, 42w)

**Comparison with images**
'I think that the way melanomas are publicised, this is what they look like, that's really misleading 'cos that isn't what mine looked like until I saw it blown up on (the dermatologist)'s screen, and I thought 'oh my God, yeah, mine does look like one of the ones on the front of the leaflet' but... it just looks very neat, symmetrical, you know, sharp edges'. (F, 39, thinner, other, 78w)

'When you go to the hospital and you see the things on the walls, and on the internet, and you see the diagrams of it, that is to me what malignant melanoma looks like. Mine didn't look... it just didn't come into the category of melanoma, it hadn't (gone) funny shaped, it hadn't been jaggy, it didn't go dark, it didn't get bigger...it just wasn't what I imagined melanoma to look like. I think of (melanoma) getting bigger, crustier, bleeding … and this was… dead flat… none of the things that were there at the back of my mind actually rang any alarm bells'. (F, 58, thinner, SSM, 22w)

'It didn't look like a melanoma. Even the booklet I've got given since… four or six pictures in there of actually different ones and it didn't look like one of them…Even like the doctor said "I've noticed it on there before but I didn't take any notice"'. (M, 36, thicker, NM, 78w/3w)

**Comparison with knowledge of other cancers**
'I think if I could feel pain and know what it was, I may be more responsive to getting it treated'. (M, 74, thinner, LMM, 303w)

---

**Table 2** Descriptions of skin changes, using the Glasgow seven-point checklist (7PCL) criteria[36] and other descriptions (dis-confirming reports in pink)

| | Way feature was described | | Thinner | | | Thicker | | |
|---|---|---|---|---|---|---|---|---|
| | | | SSM n=25 | LMM n=7 | Other n=1 | NM n=10 | SSM n=10 | LMM/ Others n=10 |
| Feature - Subgroups | Thinner | Thicker | | | | | | |
| **1 7PCL criteria** | | | | | | | | |
| **1.1. Changing size** i Cover more skin | 'it had grown, it looked bigger' (M,48,SSM,2w) | 'getting bigger, but not ultra-big, no-one noticed' (M,66, other,136w) | ● | ● | ● | ● | • | ● |
| ii Raised from skin | 'dark brown part was .. more raised' (F,40,SSM,3w) | 'vertical before it curved across at the top' [M,45,SSM,4w/ 52w), 'mushroomed out… bubbled up' [M,78,NM,8w] | • | | | • | ● | ● |
| No changing size- flat to skin | | | • | | | | | |
| **1.2. Changing and/or irregular shape** | 'it sort of made.. a pinky horseshoe' [F,53,SSM,14w/ 52w), 'maple-leaf raggedy' [F,63, LMM,104w] | 'breaking into several bits' [F,48,other,4w/78w] | ● | ● | ● | | • | • |
| No changing shape- smooth edge | | | • | | | | | |
| **1.3. Changing and/or irregular colour** | 'two colours, dark with a lighter section' [M,37, SSM,52w], 'slight discolour that got darker, black like oil' [M,67,SSM,208w] | 'red then turned black, lively-looking' [M,73,other,104w), 'several different colours' (M,82,other,3w) | ● | ● | ● | ● | ● | ● |
| No changing colour (not always darker) | | | • | | | • | • | • |
| **1.4. Oozing** i Bleeding | 'a new shaving blade would nick it but didn't bleed on its own' (M,74,LMM,303w) | 'noticed blood on the pillow' (M,66,NM,78w), 'forever bleeding and getting a scab' (M,86,SSM,52w) | • | • | | ● | ● | • |
| ii Discharge | - | 'thick oozy matter' (M,91,NM,60w) | | | | • | • | • |
| **1.5. Changing sensation** i Itch | 'when it felt itchy and I peeled like flaked off bits of it' (M,67,SSM,208w) | 'it was within a mole, just the smallest pimple, a red itchy spot (M,45,SSM,4w/52w) | • | | | • | • | • |
| ii Soreness | 'when caught my nail on it a little bit sore' (F,40, SSM,3w) | 'painful sort of like a wasp sting' (M,60,NM,1w) | • | | | • | • | ● |
| **1.6. Inflammation** i Texture change | 'quite bumpy' (M,63,SSM,2w/104w) | 'bubbled up' (M,78,NM,8w) | • | | | • | ● | • |
| ii Crusty, flaky | 'it was very dry, a bit scaly' (F,37,SSM,8w) | 'dark leathery, I tried to keep it moisturised' (F,48,other,4w/ 78w) | • | • | | ● | • | • |
| **1.7 Large size** | 'larger than a mole, about the size of a one penny piece' (F,66,LMM,20w/78w) 'size of a thumb nail' (F,63,LMM,104w) | 'It was like a 2p piece' (M,82,other,3w) 'felt this huge lump' (M,40,other,4w/68w) | • | • | | • | • | • |
| **2 Other descriptions** | | | | | | | | |
| **2.1 'Different'** | 'look very different from all the others' (M,63,SSM,2w/ 104w) 'not like the rest of my moles' (F,37,SSM,8w) | 'quite a big mole, nothing wrong until the spot on top' (F,54,SSM,1w), 'two were different, more livelier than the other ones' (M,73,other,104w) | • | | | | | • |
| **2.2 Small size** i Tiny/small mole Ii 'just a spot' | 'tiny, wee circular mole' (F,43,LMM, 22w) 'a little black spot, just an aging spot' (F,76,LMM,16w) | 'it was so minuscule' (M,93,LMM,22w) 'it was nothing like a mole at all, it was just like a spot' (M,64,other,20w) | ● | • | | ● | ● | • |
| **2.3 New lesion** | 'somebody else has noticed it so it must be a new one' (F,43,LMM,22w)'the mole had appeared, it was a new mole' (F,29,SSM,3w) | 'just suddenly appeared' (F,57,NM,36w) 'came very quick; it wasn't there and then it was there' (F,76,SSM,4w) | ● | ● | | • | • | ● |
| **2.4 'Always there'** | 'had been there for literally years' (M,74, SSM,14w), 'a birthmark, heart shaped, an old friend' (F,39,other,78w) | 'been there from birth' (F,56,NM,10w/16w) 'always had that mole, it didn't bother me' (F,61,SSM,26w) | ● | ● | ● | ● | ● | ● |

Reported feature per group: •=1–25%; ●=25–50%; ●=50–75%; ●=75–100%.
F, female; LMM, lentigo maligna melanomas; M, male; NM, nodular melanomas; SSM, superficial spreading melanomas; w, week.

## Patient factors
### Prioritisation choices

Many participants discussed other responsibilities in their lives which felt more important than making an appointment to consult their GP about a skin change, and therefore contributed to delays in help seeking. These competing priorities included employment, care of family members, moving house, holidays and other health concerns.

> In the cab game you can't organise things, you can't afford to be off your work. [M, 66, thicker, other, 156w]

> The six year old has got ADHD and mild autism and he's hard work, and I suppose [you're] concentrating on him most of your life like, and don't think about yourself… [M, 36, thicker, NM, 78w/3w]

> I'd been very busy with selling a house, buying a house, all the rest of it, and of course I've patients as well to see. [F, 72, thinner, SSM, 8w]

> I had an ulcer on my leg, and redressing that, so I think I was more taken up with that getting healed… [F, 76, thicker, NM, 4w (Community nurse contacted GP)]

Some people mentioned repeatedly failing to make an appointment with a HCP either because of the competing responsibilities or because a skin change was 'not a priority':

> I was supposed to have phoned up, but I forgot because it was busy at work and… it just skipped my memory. [M, 59, thinner, SSM, 1w]

> I didn't class it as an emergency…I didn't think it was important enough… [M, 64, thicker, other, 20w]

We found no differences in prioritisation choices between people with thinner and thicker melanomas.

### Influence of other family members and the social network

Many participants had been encouraged by other people to seek medical help, either by an observation about the skin change itself, or an encouragement to make an appointment with their GP (see box 2). Some participants had not been aware of their skin change until it was noticed by another person; others had known, and were also often aware that it was continuing to change, but they were ultimately encouraged to seek help by others. The other people included family members, friends, work colleagues and people providing treatments such as beauty therapists and hairdressers. The promotion of help seeking, whether by family members or friends, did not appear to affect TTP overall, but may have acted as a trigger for many people. A few people were wrongly reassured by family members or friends that their skin change was not potentially serious. This appeared to delay timely help seeking. There was no evident difference in the influence of family members between people diagnosed with thinner or thicker melanomas.

### Triggers for help seeking

The main difference between participants with thinner and thicker melanomas was apparent with the 'triggers' that people described as they moved from the appraisal to help-seeking interval, when they realised that they 'had a reason to discuss their skin change with a, HCP' (see figure 1). While most people from both groups consulted family or their wider social network for endorsement to seek help about aspects of skin changes (changing colour, texture and size), some people with thinner melanomas also reported a heightened awareness of cancer from family experiences or the non-medical media, or noticing their skin changes as 'different to normal', while participants with thicker melanomas appeared to depend on prompts such as the more 'red flag' symptom of oozing/bleeding.

---

**Box 2** Illustrative quotations of the influences of comments from extended family and friends

**Noticing a skin change**

'My partner's daughter says to me 'Have you always had that mole on your ear?' So I thought well somebody else has noticed it so it must be a new one'. *(F, 43, thinner, LMM, 22w)*

'I saw my sisters at the funeral and they both mentioned it; we hadn't noticed it'. *(M, 66, thicker, NM, 78w)*

**Acting on encouragement to seek help**

'It wasn't till my daughter-in-law come over from Australia, and she said to me that she thought I ought to have it checked out because obviously in Australia they're very conscious of it all'. *(F, 58, thinner, SSM, 22w)*

'The girl in the beauty salon… she always asked me about this one… and then I went again and she said, 'have you seen a doctor?', I said, 'no, I don't because it's nothing, I feel okay', and she said, 'no, please, I will make you a cup of tea, I will give you a phone number, please go this week'. *(F, 40, thinner, SSM, 156w)*

'I was at a dinner with my daughter, and fortunately I had a low backed dress on, and one of her friends said, "I don't like the look of that mole on your back, and I suggest you have it checked out"'. *(F, 66, thicker, SSM, 4w)*

'It dinnae change dramatically, so one day I quietly said to nurse friend, "Will you just have a look at this for me?" She just took one look and she says "You must promise me when you get home you will go and see the doctor"'. *(F, 61, thicker, SSM, 26w)*

**Advice of others having little effect**

'She thought it was getting darker at some stage, can't remember exactly when, but she maybe nagged me for a year or two before'. *(M, 67, thinner, SSM, 208w)*

'It's not as if I hadnae been told to go and see about it, because my daughter and my wife… they said, 'Well you should go and see about that,' but I never did, you know, until May'. *(M, 64, thicker, other, 20w)*

**Not encouraging help-seeking**

'Fairly early on I discussed it with (a friend)… But because she said, "I can't really feel it" I think I ignored it. It would have been better if she'd said to me, "I think you need to have it looked at." I think I'd have gone to the doctor then, but because she said, "No, I think it's fine" I think I left it'. *(F, 64, thicker, LMM, 104 w)*

Table 3  Time from first detecting a skin change to first presentation, and first to second presentations to primary care, by time intervals (ordered by first TTP), gender, age, melanoma type and stage

| | Time from detecting a skin change to first presentation | Time from first presentation to second presentation | Gender and age | Type and stage |
|---|---|---|---|---|
| Thinner melanomas | | | | |
| 1 | 2w | 104w | M, 63 | SSM, IA |
| 2 | 4w | 22w | F, 58 | SSM, IA |
| 3 | 14w | 52w | F, 53 | SSM, IA |
| 4 | 20w | 78w | F, 66 | LMM, IA |
| Thicker melanomas | | | | |
| 5 | 1w | 17w | M, 75 | SSM, IIC |
| 6 | 3w | 1w | M, 73 | NM, IIB |
| 7 | 4w | 52w | M, 45 | SSM, IIA |
| 8 | 4w | 68w | M, 40 | Other, IIIA |
| 9 | 4w | 78w | F, 48 | Acral, IIIA |
| 10 | 10w | 16w | F, 56 | NM, IIIA |
| 11 | 78w | 3w | M, 36 | NM, IIA |

F, female; LMM, lentigo maligna melanomas; M, male; NM, nodular melanomas; SSM, superficial spreading melanomas; w, week.

It was a black mole and most of my moles are dark or light brown so it was a different colour. [F, 29, thinner, SSM, 3w]

I'd seen something on that Embarrassing Bodies programme, and they did a thing about moles and what was not right and so I suppose I saw that and that sort of made me think, maybe I should go and get it looked at. [F, 54, thicker, SSM, 1w]

It started to bleed, that was the point at which I went to the doctor 'cos I thought it shouldn't be bleeding. [F, 64, thicker, LMM, 104w]

## Healthcare providers and system factors

Issues concerning healthcare providers and the NHS were only mentioned by a minority of participants. The first and most important area of concern involved a group of participants (n=11; thinner=4 (SSM 3, LMM 1); thicker=7 (SSM 2, NM 3, acral 1, blue malignant naevus 1) who reported that they had previously shown their lesion to an HCP, and had been reassured that they did not need further treatment (see table 3). While some just made a passing reference to their first, reassuring encounter with their GP, others gave far more detailed descriptions. A first encounter often appeared to delay a second visit to the GP by providing 'false reassurance' about the lesion. Some mentioned that they had not been adviced (oral, written or a website) on how to best monitor their lesion and what changes should alert them to return to their GP; this could potentially result in thicker lesions at diagnosis.

When people have told you that it's okay… I sort of took me eye off the ball really because I thought, well, they know better than I do. [M, 75, thicker, SSM, 1w/17 w]

Some people had problems with accessing their general practice for an appointment and, for a few busy people, this problem was exacerbated by having competing priorities.

Trying to get an appointment with the GP here can just be horrific and because I'm out on the road… I have to plan these things a couple of weeks ahead. [F, 40, thinner, SSM, 3w]

A few people with thicker melanomas also mentioned a dislike of seeing doctors, either in general practices or hospitals, so this might have delayed help seeking.

I'm just not a hospital person or a doctor person. If I'm really ill, I ken I'll have to go, but I have to be that way. [M, 52, thicker, SSM, 42w]

Patients' concerns about 'wasting their GP's time' are well known, but this concern appeared to be exacerbated by the small size of the skin changes and the lack of pain or other features that could signify more serious conditions. Again, this concern appeared more prevalent among people with thicker than thinner melanomas.

My decisions on going to the GP are always influenced to some extent by a knowledge of how busy they are and not wanting to waste their time. [F, 48, thicker, other, 4w/78w]

I think most people that I know would be afraid of the doctor saying to himself or herself, you know, there's people just coming for nothing at all. [M, 93, thicker, LMM, 22w]

A 56-year-old woman diagnosed with a stage IIIA nodular melanoma on her lower leg described her pathway over 6 months as follows:

Appraisal (10w): I've always had a mole on my leg… it was there from birth… it never bothered me because it was just flat and dark brown… It was possibly about six months ago I noticed it was just a little bit raised when it had always been flat… as if like maybe something was stuck in there…

Help seeking: I was due to have a smear and... I asked the nurse to look at it... and she said, Oh no, there's nothing to worry about, that's... I can tell these things, so she just sort of put my mind at rest... I thought, Well, she knows what she's talking about.

Re-appraisal 16 weeks: It just started obviously getting bigger and bigger. What was the worst was every time I knocked it, it bled... like a tick on you, because it was big and bulbous.

Help seeking: I realised it was getting bigger and my friend and I had talked about it and [I returned to the surgery]... it was a different nurse more senior, I have known her for years... she sort of panicked me...saying... I need to get that looked at straight away.

## DISCUSSION
### Main findings
This is the first study of detailed patient descriptions of their symptom experience and pathways to diagnosis of thinner and thicker melanomas in the UK. Addressing the policy agenda to diagnose melanoma earlier, the findings provide a number of novel insights suggesting where future interventions may be targeted. The key finding is that there appear to be subtly different patterns of symptoms experienced by those with thicker and thinner melanomas. In particular, descriptions of vertical growth, bleeding, oozing and itch were features of thicker melanomas irrespective of pathological type. Furthermore, they did not appear to occur subsequent to changes in size, shape and colour, nor just be due to location on the body, for example, not all thicker lesions were nodular melanomas on the backs of older men. There was no clear distinction between TTP and melanoma thickness. It also does not appear that those with thicker melanomas have different cognitive, emotional or behavioural responses to skin changes compared to those with thinner melanomas, or have different pathways to or through the healthcare system. While help seeking was often postponed because of other life concerns, most decisions to seek help were triggered by common factors such as advice from family and friends.

We also found a mismatch between the textual information and published images currently available, and the skin changes that were noticed by our participants. This provides opportunities to incorporate more appropriate lay vocabulary and photographic images into targeted community NHS and charity-run awareness campaigns such as 'Be Clear on Cancer' and 'Detect Cancer Early'.[26] A small but important minority of participants did not have their developing melanomas recognised during their first primary care consultation, and were not provided with enough information about on-going assessment of further skin changes or when to return to their clinician. These 'safety-netting' opportunities could be improved by more systematic approaches by GPs.

### Strengths and weaknesses
Our methodological approaches have a number of strengths. We do not know of any other studies worldwide which have compared the patient experience across the appraisal and help-seeking intervals between people with thinner and thicker melanomas. We recruited participants systematically from dermatology clinics in two contrasting regions over 12 months, and interviewed all the consenting patients diagnosed with the much less common melanomas ≥ 2 mm thickness with a poorer prognosis. The thicker melanoma group included equal numbers of NMs, SSMs and other rare and unclassified types, although no amelanotic lesions; the diversity of types in this group suggest that the differences identified between the thinner and thicker groups cannot simply be considered due to the biological differences between SSMs and NMs. Furthermore, using semi-structured interviews soon after diagnosis reduced recall bias and allowed participants to speak freely about the period leading up to their diagnosis.

We used novel and rigorous approaches to data collection, with the Model of Pathways to Treatment to underpin the interview schedule, and four different data collection methods including the use of patient drawings. Asking people to draw their skin changes and developing melanomas was of value to a number of participants, allowing them to describe subtle changes in more detail, and also to corroborate the accuracy of their recall of timing and events. Calendar landmarking was also of value to a large minority of participants, who were able to refine their recall of events and time intervals along their TTP.[19] Data saturation was reached before the total sample had been interviewed, suggesting that our findings are robust and representative of people diagnosed with melanoma in these regions of England and Scotland. As recommended in a 2006 review of symptom interpretation as a source of delay in melanoma presentation,[27] we increased the rigour of our research by applying a theoretical approach (the Pathways to Treatment model[17 18]) to frame our data collection and analysis. We conducted and reported this study according to the Aarhus statement guidelines on early cancer diagnosis research.[22]

The main weakness is that the interviews are necessarily retrospective and subject to recall and framing bias. As a result, the accounts cannot be regarded as an exact description of what happened. Instead, they are narratives that allowed people to describe their experiences and reflect a post hoc rationalisation of events framed by their subsequent encounters with HCPs and increased knowledge since the diagnosis. Although we recruited all patients with thicker melanomas compared with purposive recruitment for thinner melanomas we believe the groups were similar as the latter groups were matched for gender, age, geographic location and season. Furthermore, people from these two UK regions may have different beliefs and experiences of the pathway to melanoma diagnosis from people in other

UK regions, and patients who did not agree to take part in the study may have affected the representativeness of the sample.

## Comparison with the existing literature

While there is a paucity of qualitative studies undertaken with people soon after their melanoma diagnosis, our findings resonate with a grounded theory study undertaken in northern England that explored the meaning to people treated for melanoma of shorter and longer time lapses between detecting signs and receiving treatment,[28] and those from an interview study about factors influencing presentation in primary care, undertaken with patients with suspicious pigmented lesions (only 4/40 interviewees were later diagnosed with melanoma).[15] A French questionnaire study set among 590 people with melanomas also showed that relatives were involved in the detection of half of the melanomas, with median delays of 4 months before the patient realised they had a suspicious lesion, and further median delays of 2 months before this lesion was seen by a doctor.[29] Other evidence around time to diagnosis, but not comparing thinner and thicker melanomas, comes mainly from retrospective review of medical records or dermatologist experience, and suggests similar times to diagnosis and diagnosis.[30]

## Implications for clinicians and policymakers

Policymakers continue to face the challenge of a widespread lack of awareness of cancer symptoms among the UK general population,[31] and that there are the significant barriers to help seeking.[32] Policy responses have included campaigns to raise symptom awareness, with major investment in the new 'Be Clear on Cancer' skin cancer pilot campaign in SW England. Our findings clearly demonstrate that the words and images in current use may not meet the needs of the population who are likely to be assessing their skin changes at an early stage in tumour development. Current images tend to represent more extreme changes which may not always be present. Future melanoma awareness campaigns, as well as NHS and charity websites giving information about skin checks, would be advised to provide more evidence around the features of early skin changes using lay vocabulary,[33] to consider their selection of images of early melanomas for a better 'match' with people's observations, and to provide more evidence around prompts to encourage timely help seeking. They should also consider more targeted approaches such as focusing on: providing higher risk groups such as older men with tailored information, lay vocabulary and images; giving families and friends advice on how to check each other's skin regularly; and talking with professional groups from the hair, beauty, and exercise industries who also undertake informal skin checks.

Several participants reported visiting their GP or other HCP on more than one occasion and some were given false reassurance. The average GP working in the UK will only diagnose a melanoma every 2–3 years but will commonly be consulted about a pigmented skin lesion, often after other health issues have already been discussed in the consultation. While we recognise the challenges facing GPs when differentiating potentially rare and serious conditions such as melanoma from common and benign conditions, this study suggests that some patients are not being provided with adequate information either about monitoring their skin changes or what changes should prompt another consultation. The principles of 'safety-netting' have been disseminated by the RCGP and could be applied more effectively; they include recommendations for appropriate advice and written information for patients about the warning symptoms, monitoring symptoms, when to make a follow-up appointment, and reassurance to patients that symptoms like skin changes warrant GP attention, thus 'legitimising' a follow-up visit.[34]

## Unanswered questions and future research

While the findings of this qualitative study are of immediate importance to primary care clinicians and policymakers, there are also suggestions of subtly different patterns of symptoms experienced by those with thicker and thinner melanomas, irrespective of pathological type. The descriptions of vertical as well as horizontal growth, and bleeding, oozing or itch were particular features of thicker melanomas but not only NMs. Furthermore, they did not appear to occur subsequent to changes in size, shape or colour, so may not necessarily be later features of melanoma. Although these symptom clusters may be more related to tumour biology than differences in symptom appraisal and help seeking, these interesting differences need further exploration with bigger and more diverse populations and quantitative as well as qualitative study designs.

Alternative approaches to raising symptom awareness and supporting monitoring of skin changes to prompt earlier help seeking may be needed. There is a growing interest in the application of smartphone technology as one such approach, and it is clearly an area for further research,[35] but concerns remain around the safety and utility of this technology.

**Author affiliations**
[1]The Primary Care Unit, University of Cambridge, Cambridge, UK
[2]General Practice and Primary Care Academic Centre, University of Melbourne, Melbourne, Australia
[3]Centre for Population Health Sciences, University of Edinburgh, Edinburgh, UK
[4]Unit of Social & Behavioural Sciences, Kings College London Dental Institute, London, UK
[5]Cambridge University Hospitals NHS Trust, Cambridge, UK
[6]Royal Infirmary of Edinburgh, NHS Lothian, Edinburgh, UK
[7]University of Glasgow, Glasgow, UK

**Acknowledgements** The authors particularly thank all the patients who kindly gave up their time and shared their personal accounts with us. They also thank Vicky McMorran and Sheena Dryden, melanoma/skin cancer nurse specialists at Cambridge University Hospitals NHS Foundation Trust and the

Edinburgh Royal Infirmary NHS Lothian, respectively, for their enthusiastic help with recruitment. The authors are also grateful to our two patients who gave us insights and feedback throughout the study, to Anna Barford for her early contribution to study set-up and data collection, to James Brimicombe for advice on data management and for developing the research database, and to Mr Per Hall for his support and encouragement throughout the study and comments on the final manuscript.

**Contributors** This study arose from collaboration between members of the National Cancer Research Institute (NCRI) Primary Care and Melanoma Clinical Studies Groups. FMW, CC, RM, SS and DW were involved in the design of the study. LB and DC performed all the interviews and led the analysis, contributing to the core study team together with FMW, SS and CC. FMW wrote the first draft of the manuscript; all authors reviewed and edited the manuscript.

**Funding** Thanks to our funding organisation the National Awareness and Early Diagnosis Initiative (NAEDI) (Project Award C8640/A12226), and to their funding partners: Cancer Research UK; Department of Health, England; Economic and Social Research Council; Health and Social Care Research and Development Division; Public Health Agency, Northern Ireland, National Institute for Social Care and Health Research, Wales and the Scottish Government. The views expressed in this publication are those of the author (s) and not necessarily those of the NHS, the National Institute for Health Research or the Department of Health.

**Competing interests** FMW was supported by an NIHR Clinical Lectureship followed by a NIHR Clinician Scientist award at the time of this study.

**Patient consent** Obtained.

**Ethics approval** The study obtained ethical approval from the Cambridgeshire 4 Research Ethics Committee (11/EE/0076). The study was CKCRN approved (ID number 10310), and obtained NHS governance approvals from Cambridge University Hospitals NHS Foundation Trust's Research & Development Department, and NHS Lothian (Lothian R&D Project No: 2011/R/DER/04).

**Provenance and peer review** Not commissioned; externally peer reviewed.

**Data sharing statement** No additional data are available.

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
