## [Reviewer comments · BMJ Open]

Some articles will have been accepted based in part or entirely on reviews undertaken for other BMJ Group journals. These will be reproduced where possible.

ARTICLE DETAILS

TITLE (PROVISIONAL)	'This isn't what mine looked like': a qualitative study of symptom appraisal and help seeking in people recently diagnosed with melanoma
AUTHORS	Walter, Fiona; Birt, Linda; Cavers, Debbie; Scott, Suzanne; Emery, Jon; Burrows, Nigel; Kavanagh, Gina; MacKie, Rona; Weller, David; Campbell, Christine

VERSION 1 - REVIEW

REVIEWER	Pippa Corrie Cambridge University Hospitals NHS Foundation Trust I am a member of the same specialist MDT as one of the co-authors, Dr N Burrows. I am a steering committee member for the 'MelaTools' research study for which Dr Fiona Walter is CI
REVIEW RETURNED	12-May-2014

GENERAL COMMENTS	The authors present interesting findings after interviewing a cohort of 63 patients diagnosed with either a 'thinner' (<1mm) or 'thicker' (>2mm) melanoma. The paper is well written and informative, but there are a few points to raise. 1. Abstract conclusions I think the first statement needs clarification, since the patients clearly did ultimately recognise concerns about their skin lesion sufficient to seek medical attention. 2. p3 Strength and limitations of the this study and p4 introduction line 35 make reference to thicker, T3 &T4 melanomas having 5 year disease free prospects of <45% and the CRUK melanoma webpage, risks-and-benefits is cited. I could not find this information there, but in a separate webpage: stats-and-outlook, 5 year survival rates are given by disease stage. It is important to provide accurate outcome predictions since T3 and T4 melanomas in the absence of lymph node involvement have a higher predicted 5 year survival rate than 45% overall. Tumours >4mm certainly do worse, but this study looked at patients with tumours >2mm and I could not find anywhere the number of patients whose tumours were >4mm. If the authors do have disease-free as opposed to overall survival data to justify the 45% quoted, it should be accurately referenced. Table 1: Please include median Breslow thickness and range for both groups. The readership I suspect will not have detailed knowledge of melanoma TNM staging. I think it would be more informative to breakdown the melanoma TNM stage categories to TXNXMX and provide a footnote to explain the categories of T and
--

	N.
--	----

REVIEWER	Peter Murchie University of Aberdeen, Centre for Academic Primary Care United Kingdom
REVIEW RETURNED	14-May-2014

GENERAL COMMENTS	OVERALL Thank you for asking me to review this very well conducted and written qualitative paper. It is also fascinating and valuable. It presents a rigorous qualitative study which explores symptom appraisal and help seeking in people recently diagnosed with melanoma with respect to their own characteristics and the characteristics of their cancer. There are some limitations in presentation, methodology and limitation but these are largely acknowledged. The data are, as the authors say, of considerable value to those tasked with researching and implementing way to expedite melanoma diagnosis. I only have few concerns with the paper – detailed below – which should be readily addressed. As it is a qualitative paper and quite long already I do think that the authors could generally shorten the introduction, methods and discussion sections. ABSTRACT Could the results and conclusions be conveyed a little more succinctly, perhaps as bullet points? INTRODUCTION I think the introduction is too long. The first paragraph relating to NAEDI etc is almost redundant and could be deleted without detriment. The second paragraph is almost redundant when really all this is required is a succinct expression of the fact that melanoma is increasing in Caucasian populations. METHODS In the setting and recruitment section it would be useful to know how many patients were excluded as unsuitable and briefly why. Also, perhaps there could be some brief reflection in the discussion on whether the inclusive versus purposive recruitment for thick and thin melanomas mattered. Were the think group demographically different for example? Using the Models of Pathways to Treatment to underpin the interview schedule is a key strength of the study. Similarly, I am impressed by the use of four different data collection methods (schedule, calendar, diaries, drawings). This, especially the use of patient drawings, should be highlighted as a novel and rigorous approach. Another strength is that the interviews were all conducted within 10 weeks. How data saturation was defined and decided needs to be mentioned in the methods. It is currently not raised until the discussion. The analysis also mentions 17 pilot interviews. It is not clear if these
--

	were included or additional to the 63 patients interviewed for the study. If the latter perhaps a little more information is needed about the pilot exercise. In the analysis section the second last paragraph contains data about patients with shorter intervals tending to use diaries and difficulties that patients has recalling triggers to consulting. These are really more suited to the results section. The cross comparison of data from different patient groups is another strength of this paper. RESULTS Are the authors able to offer any insight into why almost 50% of those approached declined to take part. The paragraph on patient characteristics could be abbreviated and closer reference made to the table. The authors quote a range of 1-303 weeks. 303 seems very long indeed if the patient truly had a melanoma. How many patients reported delays of more than , say a year. The paper might benefit on some consideration of distinguishing patients who were overstating their delay – as this seems possible with a delay of almost 6 years! The results section is extremely well written and is fascinating to read. DISCUSSION I think a limitation of the paper that is not really considered relates to the representativeness of the sample interviewed to this population of melanoma patients and the UK population as a whole. I think the 50% decline rate makes this important. The authors present a very good argument for the import of their data for clinicians and policy makers with which I agree.
--	---

VERSION 1 – AUTHOR RESPONSE

Reviewer (1) Pippa Corrie

The authors present interesting findings after interviewing a cohort of 63 patients diagnosed with either a 'thinner' (<1mm) or 'thicker' (>2mm) melanoma. The paper is well written and informative, but there are a few points to raise.

1. Abstract conclusions

I think the first statement needs clarification, since the patients clearly did ultimately recognise concerns about their skin lesion sufficient to seek medical attention.

We agree that the patients did ultimately recognise concerns about their skin lesions sufficient to seek medical attention but our first statement was making the point that if they had recognised skin changes sooner they may have sought more timely medical attention. This may have led to diagnosis with a thinner lesion requiring less radical treatment and with a better prognosis. We would therefore

prefer to retain the sentence as:

Patients diagnosed with both thinner and thicker melanomas often did not initially recognise or interpret their skin changes as warning signs or prompts to seek timely medical attention.

2. p3 Strength and limitations of the this study and p4 introduction line 35 make reference to thicker, T3 & T4 melanomas having 5 year disease free prospects of <45% and the CRUK melanoma webpage, risks-and-benefits is cited. I could not find this information there, but in a separate webpage: stats-and-outlook, 5 year survival rates are given by disease stage. It is important to provide accurate outcome predictions since T3 and T4 melanomas in the absence of lymph node involvement have a higher predicted 5 year survival rate than 45% overall. Tumours >4mm certainly do worse, but this study looked at patients with tumours >2mm and I could not find anywhere the number of patients whose tumours were >4mm. If the authors do have disease-free as opposed to overall survival data to justify the 45% quoted, it should be accurately referenced.

We thank the reviewer for this helpful clarification. We have amended the manuscript as follows:

- P 3: This study is the first exploration of symptom appraisal and help-seeking among people diagnosed with 'thinner' melanomas (T1, very good prognosis, 5 year disease-free prospects 95%), compared with those with 'thicker' melanomas (T3 and T4, less good prognosis, 5 year disease-free prospects <55%).
- P4: Patients with a primary melanoma ≤ 1 mm at diagnosis (T1) currently have 5 year disease-free prospects of over 95%, while for tumours ≥ 2 mm at diagnosis this is lower, falling to <55% with lymph node involvement but no metastatic spread [10].
- Reference: Cancer Research UK.

<http://www.cancerresearchuk.org/cancerinfo/cancerstats/types/skin/survival/#stage> (accessed 8 June 2014).

Table 1: Please include median Breslow thickness and range for both groups.

These data have been included in the Results section:

Table 1 shows the demographic and self-reported skin characteristics of the 63 study participants, and the clinical characteristics of their melanomas, comparing participants with thinner (n=33, median Breslow thickness 0.5 mm, range 0.1-0.9 mm) and thicker (n=30, median Breslow thickness 3.5 mm, range 2.1-12.0 mm) melanomas.

The readership I suspect will not have detailed knowledge of melanoma TNM staging. I think it would be more informative to breakdown the melanoma TNM stage categories to TXNXMX and provide a footnote to explain the categories of T and N.

As suggested, we have added TXNXMX stage categories to the footnotes to Table 1:

d Stage IA = 27, stage IB = 6 (T1-2a, N0, M0).

e Stage IIA = 11, stage IIB = 5, Stage IIC = 7 (T2b-4b, N0, M0).

f Stage IIIA = 6, stage IIIB = 1 (T1a – 4a, N1a-2c, M0).

Reviewer (2) Peter Murchie

OVERALL

Thank you for asking me to review this very well conducted and written qualitative paper. It is also fascinating and valuable. It presents a rigorous qualitative study which explores symptom appraisal and help seeking in people recently diagnosed with melanoma with respect to their own characteristics and the characteristics of their cancer. There are some limitations in presentation, methodology and limitation but these are largely acknowledged. The data are, as the authors say, of

considerable value to those tasked with researching and implementing way to expedite melanoma diagnosis.

Thank you for these very supportive comments.

I only have few concerns with the paper – detailed below – which should be readily addressed. As it is a qualitative paper and quite long already I do think that the authors could generally shorten the introduction, methods and discussion sections.

This has been undertaken in all three sections.

ABSTRACT

Could the results and conclusions be conveyed a little more succinctly, perhaps as bullet points?

We do not feel that bullet points are appropriate for this abstract; instead, we have précised the text to make it more succinct.

INTRODUCTION

I think the introduction is too long. The first paragraph relating to NAEDI etc is almost redundant and could be deleted without detriment. The second paragraph is almost redundant when really all this is required is a succinct expression of the fact that melanoma is increasing in Caucasian populations.

The critique is helpful. We have deleted much of the first and second paragraphs for brevity and succinctness, and combined the remainder into a single paragraph.

METHODS

In the setting and recruitment section it would be useful to know how many patients were excluded as unsuitable and briefly why.

We have added the following sentence to the end of the section:

Reasons for not selecting patients for interview included: sampling decisions (n=34), lost to follow-up (n=6), and ill-health (n=1).

Also, perhaps there could be some brief reflection in the discussion on whether the inclusive versus purposive recruitment for thick and thin melanomas mattered. Were the thin group demographically different for example?

We have included a sentence in the Discussion:

Although we recruited all patients with thicker melanomas compared with purposive recruitment for thinner melanomas we believe the groups were similar as the latter group were matched for gender, age, geographic location and season.

Using the Models of Pathways to Treatment to underpin the interview schedule is a key strength of the study. Similarly, I am impressed by the use of four different data collection methods (schedule, calendar, diaries, drawings). This, especially the use of patient drawings, should be highlighted as a novel and rigorous approach.

We have added a sentence to the Discussion, and rearranged the subsequent sentences:

We used novel and rigorous approaches to data collection, with the Model of Pathways to Treatment to underpin the interview schedule, and four different data collection methods including the use of patient drawings. Asking people to draw their skin changes and developing melanomas was of value to a number of participants, allowing them to describe subtle changes in more detail, and also to

corroborate the accuracy of their recall of timing and events. Calendar-landmarking was also of value to a large minority of participants, who were able to refine their recall of events and time intervals along their time to presentation.

Another strength is that the interviews were all conducted within 10 weeks. How data saturation was defined and decided needs to be mentioned in the methods. It is currently not raised until the discussion.

We believe that most readers will understand the concept of data saturation and have therefore not added any further definition after mentioning 'and we continued until saturation of data' in the Methods section.

The analysis also mentions 17 pilot interviews. It is not clear if these were included or additional to the 63 patients interviewed for the study. If the latter perhaps a little more information is needed about the pilot exercise.

More information has been included in the Methods section:

...and a pilot study (n=17, conducted during the early stages of the study, and including patients interviewed >10 weeks post-diagnosis (n=12), or with melanoma histology which did not fit the inclusion criteria (n=5, Breslow thickness 1-2mm or indeterminate).

In the analysis section the second last paragraph contains data about patients with shorter intervals tending to use diaries and difficulties that patients has recalling triggers to consulting. These are really more suited to the results section.

While we agree that these data could be included in the Methods or Results section; we have chosen to leave them in the Methods section.

The cross comparison of data from different patient groups is another strength of this paper.

RESULTS

Are the authors able to offer any insight into why almost 50% of those approached declined to take part?

We think that most researchers will agree that a 50% consent rate among people around the time of a cancer diagnosis is very reasonable. Furthermore, ethics committee guidance suggests patients do not need to provide a reason for non-participation, therefore we were unable to record why patients declined to take part. We have therefore not added any further comment.

The paragraph on patient characteristics could be abbreviated and closer reference made to the table.

The second paragraph has been abbreviated to:

While people with thinner melanomas were younger (60.5 vs 66.1 years), the groups were otherwise similar for socio-demographic factors.

As the details concerning the histology of the melanomas are considered important by our dermatology colleagues, these are retained in the text as well as Table 1.

The authors quote a range of 1-303 weeks. 303 seems very long indeed if the patient truly had a melanoma. How many patients reported delays of more than, say a year. The paper might benefit on some consideration of distinguishing patients who were overstating their delay – as this seems possible with a delay of almost 6 years!

This is a reasonable comment although the literature does report some similar very long diagnostic delays. We have added details of the number of people with time to presentation >52 weeks to the Duration of Skin Changes paragraph of the Results section:

The time to presentation (TTP) was between 1 week and 303 weeks (thinner: median TTP 21 weeks, range 1-303 weeks, 5 longer than 52 weeks; thicker: median TTP 19 weeks, range 1-156 weeks, 7 longer than 52 weeks).

The results section is extremely well written and is fascinating to read.

DISCUSSION

I think a limitation of the paper that is not really considered relates to the representativeness of the sample interviewed to this population of melanoma patients and the UK population as a whole. I think the 50% decline rate makes this important.

A sentence in the Discussion section has been amended to:

Furthermore, people from these two UK regions may have different beliefs and experiences of the pathway to melanoma diagnosis from people in other UK regions, and patients who did not agree to take part in the study may have affected the representativeness of the sample.

The authors present a very good argument for the import of their data for clinicians and policy makers with which I agree.